# Coherent coupling between vortex bound states and magnetic impurities in 2D layered superconductors

Sunghun Park [1,7], Víctor Barrena [2,7], Samuel Mañas-Valero [3], José J. Baldoví [3,4], Antón Fente [2], Edwin Herrera [2], Federico Mompeán [5], Mar García-Hernández [5], Ángel Rubio [4,6], Eugenio Coronado [3], Isabel Guillamón [2], Alfredo Levy Yeyati [1] & Hermann Suderow [2✉]

Bound states in superconductors are expected to exhibit a spatially resolved electron-hole asymmetry which is the hallmark of their quantum nature. This asymmetry manifests as oscillations at the Fermi wavelength, which is usually tiny and thus washed out by thermal broadening or by scattering at defects. Here we demonstrate theoretically and confirm experimentally that, when coupled to magnetic impurities, bound states in a vortex core exhibit an emergent axial electron-hole asymmetry on a much longer scale, set by the coherence length. We study vortices in 2H-NbSe$_2$ and in 2H-NbSe$_{1.8}$S$_{0.2}$ with magnetic impurities, characterizing these with detailed Hubbard-corrected density functional calculations. We find that the induced electron-hole imbalance depends on the band character of the superconducting material. Our results show that coupling between quantum bound states in superconductors is remarkably robust and has a strong influence in tunneling measurements.

[1] Departamento de Física Teórica de la Materia Condensada, Instituto Nicolás Cabrera and Condensed Matter Physics Center (IFIMAC), Universidad Autónoma de Madrid, Madrid, Spain. [2] Laboratorio de Bajas Temperaturas y Altos Campos Magnéticos, Departamento de Física de la Materia Condensada, Instituto Nicolás Cabrera and Condensed Matter Physics Center (IFIMAC), Unidad Asociada UAM-CSIC, Universidad Autónoma de Madrid, Madrid, Spain. [3] Instituto de Ciencia Molecular (ICMol), Universidad de Valencia, Paterna, Spain. [4] Max Planck Institute for the Structure and Dynamics of Matter, Hamburg, Germany. [5] Instituto de Ciencia de Materiales de Madrid, Consejo Superior de Investigaciones Científicas (ICMM-CSIC), Madrid, Spain. [6] Nano-Bio Spectroscopy Group and European Theoretical Spectroscopy Facility (ETSF), Universidad del País Vasco CFM CSIC-UPV/EHU-MPC & DIPC, San Sebastián, Spain. [7] These authors contributed equally: Sunghun Park, Víctor Barrena. ✉email: hermann.suderow@uam.es

**B**ound states appear in superconductors at localized perturbations in the superconducting order parameter. Both Caroli de Gennes Matricon (CdGM) states at vortex cores[1,2] and Yu-Shiba-Rusinov (YSR) states at magnetic impurities[3–5] are examples of this phenomenon. YSR states provide mixed electron-hole excitations that serve to create the conditions needed for Majorana states. These are expected for instance at the ends of magnetic chains of YSR atoms[6–9]. On the other hand, CdGM states have been proposed to isolate and manipulate Majoranas in a topological superconductor[10–15].

The nature of YSR and CdGM states is, however, quite different. YSR states are spin polarized and appear at a single or a few subgap energies and exhibit oscillations at the Fermi wavelength $\lambda_F$ that can be resolved with atomic scale local density of states (LDOS) measurements[16–19]. By contrast, CdGM states are spin degenerate and form a quasi-continuum with a level separation $\Delta^2/E_F$ (where $\Delta$ is the superconducting gap and $E_F$ is the Fermi energy), which is usually small compared to $\Delta$. Thus, their discreteness and their mixed electron-hole character only appears at very low temperatures or for $\Delta \approx E_F$ and in absence of scattering, in the so called quantum limit[20]. Otherwise, thermal excitations or defects produce dephasing resulting in an electron-hole symmetric LDOS pattern at vortex cores.

Thus, in most cases, CdGM states are electron-hole symmetric and their features in the LDOS extend to much larger distances than those of YSR states. Here, we ask the question if we can build a hybrid quantum system consisting in vortices close to magnetic impurities and transfer the quantum property of YSR states, i.e. their electron-hole asymmetry, into the more extended CdGM states far from the quantum limit.

As we show below, we indeed theoretically predict and experimentally observe electron-hole asymmetric features in the LDOS of vortices in presence of magnetic impurities. As we schematically represent in Fig. 1, a magnetic impurity close to a vortex core induces a coupling between CdGM states with $n$ and $n \pm 1$ angular momenta. This coupling produces a slight shift of the charge density of the positive (negative) energy excitations towards (outwards) the impurity with respect to their mean position, which remains even away of the quantum limit. The sign of the coupling changes when the bands at the Fermi energy have a hole (electron) character. The discrete nature of CdGM

states is thus revealed in the vortex core LDOS: the difference between the electron and hole LDOS would exhibit an axial asymmetry as illustrated in Fig. 1d, e, with a larger LDOS close to the position of the magnetic impurity. We emphasize that, in contrast to the above mentioned oscillations at the tiny $\lambda_F$ scale, the electron-hole asymmetric feature in the vortex LDOS occurs at a much larger length scale.

The transition metal dichalcogenide two-dimensional (2D)-layered superconductor 2H-NbSe$_2$ is the first material where the vortex LDOS has been measured and one of the few where the nature of CdGM states has been extensively studied, both in experiment and theory[21–23]. YSR impurities have been also imaged in detail in this material[18,24,25]. Vortex cores in 2H-NbSe$_2$ ($T_c = 7.2$ K) are highly anisotropic, with a characteristic sixfold star shape. Previous work imaged YSR impurities and vortex cores at the same time, but did not identify any particular connection[18]. On the other hand, when doped with S as in 2H-NbSe$_{1.8}$S$_{0.2}$ ($T_c = 6.6$ K) the vortex core CdGM states are in-plane isotropic, leading to round shaped, symmetric, vortex cores[26]. Using these two systems, we can study the interaction between YSR and CdGM states for in-plane isotropic (2H-NbSe$_{1.8}$S$_{0.2}$) and anisotropic (2H-NbSe$_2$) vortices.

## Results

**Length scales for CdGM and YSR states.** It is of interest to first analyze the different length scales associated with isolated CdGM and YSR states, particularly in the case of 2H-NbSe$_{1.8}$S$_{0.2}$ (Fig. 2). As we show in Fig. 2a, CdGM states provide a zero bias peak at the center of the vortex that decays with distance at a scale, which is generally larger than the coherence length (of approximately 10 nm in 2H-NbSe$_2$ and 7 nm in 2H-NbSe$_{1.8}$S$_{0.2}$ as obtained from $H_{c2}(T)$) and is magnetic field dependent[26,27]. The vortex core size at the magnetic fields considered here is of $\xi_V \approx 30$ nm[26]. The zero bias peak splits when leaving the vortex core, as shown in previous work[20–23,26,28,29]. The sixfold anisotropy characteristic of vortex cores in 2H-NbSe$_2$ is washed out by the S substitutional disorder in 2H-NbSe$_{1.8}$S$_{0.2}$. On the other hand, a YSR state in 2H-NbSe$_{1.8}$S$_{0.2}$ is shown in Fig. 2b. There is a conductance peak within the superconducting gap, which changes from positive to negative bias voltage values at a scale of order of $\lambda_F$ (about 0.7 nm, see lower right inset in Fig. 2b and ref. [18]). At the same time, the

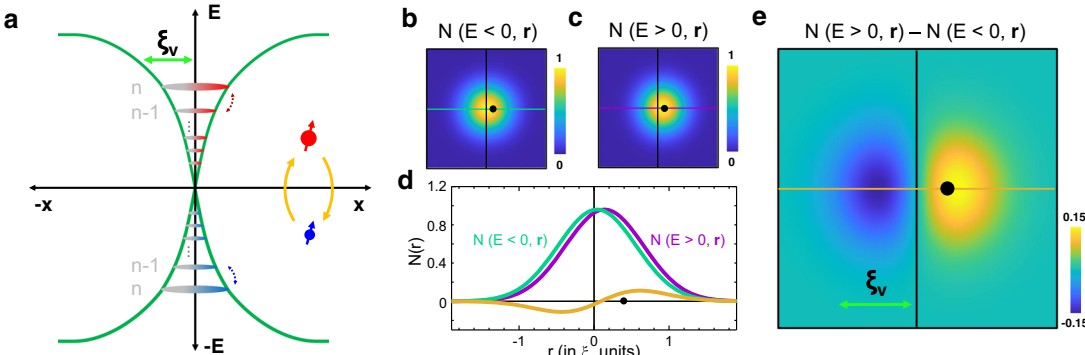

**Fig. 1 Interaction between Caroli de Gennes Matricon (CdGM) and Yu Shiba Rusinov (YSR) states. a** The YSR state is schematically represented by the red and blue dots with arrows representing the spin. The value of the superconducting gap as a function of the radius is shown in green for energies above and below the Fermi level. The CdGM states are schematically shown by colored circles. The difference in energy between levels, $\Delta^2/2E_F$, with $\Delta$ the superconducting gap and $E_F$ the Fermi energy, is strongly exaggerated. Green arrow on top schematically represents the vortex core size $\xi_V$. The interaction between CdGM and YSR bound states leads to a spatial shift in the CdGM states, which is different for electron and hole components of the Local Density of States (LDOS) inside the superconducting gap, $N(E > 0)$ (**b**) and $N(E < 0)$ (**c**). In **d** we show line profiles of the images in **b** (blue-green) and of **c** (magenta). We also show a profile of their difference in orange. In **e** we show the difference image between **b** and **c**. The LDOS is larger along the direction of the YSR impurity. This occurs for a hole band and a single impurity. For an electron band, we expect the opposite result. The position of the YSR impurity is represented by a black dot and the vortex core center by the crossing point between lines in **b**, **c**, **d**, and **e**.

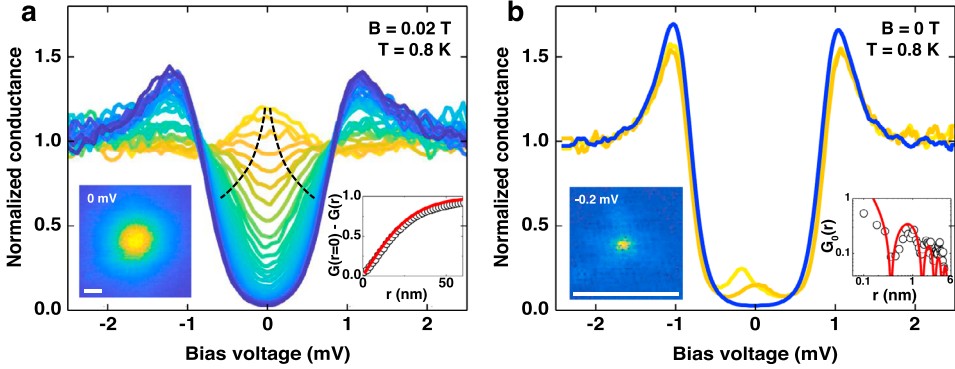

**Fig. 2 Length scales of CdGM and YSR states. a** Tunneling conductance from the center of a vortex (yellow to blue) in 2H-NbSe$_{1.8}$S$_{0.2}$. Dashed lines represent schematically the splitting of the zero bias conductance peak when leaving the vortex core, a characteristic feature of CdGM states. In the lower left inset we show a zero bias conductance map. In the lower right panel we show as white circles the radially averaged zero bias conductance $G$ as a function of the radius. **b** Tunneling conductance as a function of the position at a magnetic Fe impurity (yellow and ochre curves are close to the impurity center and separated approximately by $\lambda_F \approx 0.7$ nm) and far from it (blue) in 2H-NbSe$_{1.8}$S$_{0.2}$. The tunneling conductance map at the bias voltage where we observe a maximum of the conductance in the yellow curve is shown in the lower left panel. In the lower right panel we show as white circles the spatial dependence of the conductance along a line from the position of the impurity outwards. Red lines in the lower right insets follows roughly the expected dependencies for the spatial variation of the LDOS for CdGM (**a**) and YSR (**b**) states, discussed in previous work[18,21–23]. White bars in the lower left insets of **a**, **b** are both 10 nm long.

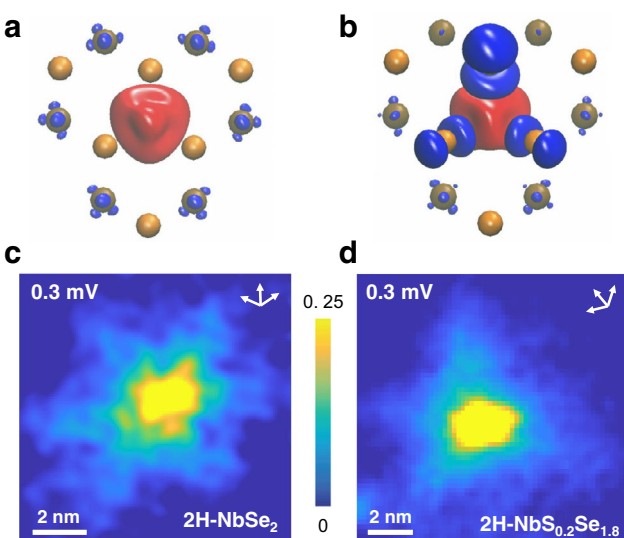

**Fig. 3 YSR states at magnetic impurities.** Spin-density isosurface of 2H-NbSe$_2$ (**a**) and of 2H-NbSe$_{1.8}$S$_{0.2}$ (**b**) obtained from DFT + U calculations. Red stands for spin up and blue for spin-down charge density. The isosurface is plotted for an imbalance by 0.002 of the spin density. The Fe atom (not shown) is located on the red spot, substituting a Nb atom. Se atoms are orange and Nb atoms ochre colors. We represent the two uppermost layers (Se and Nb) from the top. In **b** there are S atoms in layers below, shown in the Supplementary Fig. 4. Calculations are made on four layers, using 4 × 4 × 2 sized slabs (see Supplementary Note 3, for further details). **c**, **d** Measured tunneling conductance maps at a Fe impurity in 2H-NbSe$_2$ (**c**) and in 2H-NbSe$_{1.8}$S$_{0.2}$ (**d**). White arrows provide the crystalline directions of the atomic Se lattice.

height of the peak decreases exponentially with distance (lower right inset of Fig. 2b). Thus, we see that the effect of YSR states is transposed two orders of magnitude in distance, from $\lambda_F$ to $\xi_V$, and leads to the electron-hole asymmetric CdGM states we discuss below.

**Magnetism of substitutional Fe impurities**. To better understand the magnetism at the Fe impurities we calculated the electronic structure and the spin-density isosurface, i.e., the

difference between spin-up and spin-down charge densities (Fig. 3), by means of Hubbard-corrected density functional theory (DFT+U). We constructed a slab model with a 4 × 4 × 2 supercell that contains one isolated magnetic impurity at a Nb site. To model 2H-NbSe$_{1.8}$S$_{0.2}$, we introduced approximately 10% S atoms (115 Se and 13 S atoms) randomly distributed. Computational methods are detailed in the Supplementary Note 3. In 2H-NbSe$_2$ we observe clearly a strong magnetic moment on the Fe atom (Fig. 3a). The same behavior is observed at the Fe site in 2H-NbSe$_{1.8}$S$_{0.2}$ with, however, a large antiferromagnetic coupling with immediately neighboring Se atoms that become spin-polarized (Fig. 3). Importantly, this coupling breaks the sixfold in-plane symmetry of the Se lattice. The introduction of S atoms lowers the symmetry from six- to threefold. We show in Fig. 3c, d the measured tunneling conductance map $G(\mathbf{r}) = G(x,y)$ on a YSR state on 2H-NbSe$_2$ (Fig. 3c) and on 2H-NbSe$_{1.8}$S$_{0.2}$ (Fig. 3d). We see that the LDOS at YSR impurities is modified from a sixfold star shape in 2H-NbSe$_2$ to a predominantly threefold star shape in 2H-NbSe$_{1.8}$S$_{0.2}$ and ascribe this effect to the symmetry breaking in the lattice induced by the S distribution in 2H-NbSe$_{1.8}$S$_{0.2}$, as suggested by our calculations (Fig. 3a, b). The threefold symmetry is smeared at the scale of $\xi_V$, leading to the observed round vortex cores shown in Fig. 2a.

**Interplay between CdGM and YSR states: electron-hole axial asymmetry in vortices**. We show vortices in close proximity to YSR impurities in Fig. 4a, d. When we make the difference between images taken at positive and negative bias voltages, $\frac{\delta G(\mathbf{r}, V)}{G_0} = \frac{G(\mathbf{r}, V) - G(\mathbf{r}, -V)}{G_0}$ (with $G_0$ the averaged tunneling conductance for bias voltages above the gap), we observe that vortex cores are not axially symmetric (Fig. 4b, e). In contrast (as we show in detail in the Supplementary Fig. 3c), $\frac{\delta G(\mathbf{r}, V)}{G_0}$ is axially symmetric in absence of YSR impurities.

As stated above, we trace the broken axial symmetry to the interplay between vortex and YSR states. We have calculated the perturbation to a rotationally symmetric vortex induced by a magnetic impurity located close to the vortex core. As shown in the Supplementary Note 1, we start with a 2D superconductor described by a Bogoliubov-de Gennes Hamiltonian. We find discrete energy levels $E_n$ and the shape of electron and hole wave functions $\psi_n^+$, $\psi_n^-$ of CdGM vortex bound states (with $n$ the

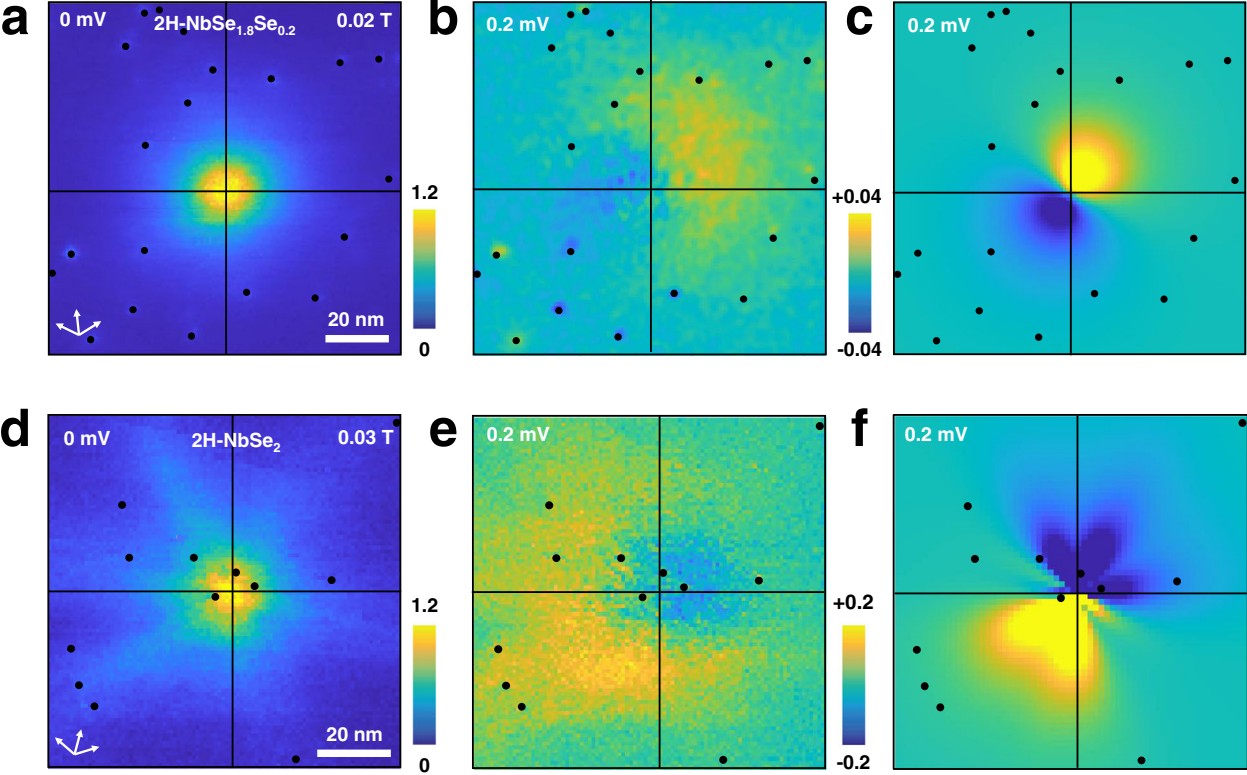

**Fig. 4 Vortex cores coupled with YSR states. a** Zero bias conductance map of a vortex in 2H-NbSe$_{1.8}$S$_{0.2}$. **b** Map showing the difference between the normalized tunneling conductance at positive and negative bias voltages $\frac{\delta G(\mathbf{r},V)}{G_0} = \frac{G(\mathbf{r},V)-G(\mathbf{r},-V)}{G_0}$ at $|V| = 0.2$ mV and in the same field of view as **a. c** $\frac{\delta G(\mathbf{r},V)}{G_0}$ obtained from the calculation described in the text. The same quantities are plotted for 2H-NbSe$_2$ in **d, e,** and **f.** Color scales are given by the bars on the right side of panels **a** and **d** for the zero bias conductance, and in between panels **b, c** and **e, f** for $\frac{\delta G(\mathbf{r},V)}{G_0}$. Fe impurities are marked by black dots. Vortex centers are at the crossing point between the black lines. White arrows in **a, b** give the directions of the Se lattice.

angular momentum number). Magnetic YSR impurities are characterized as usual by the exchange coupling $J$ at the impurity sites[30]. This coupling leads to an effective Hamiltonian in the subspace spanned by the states $\psi_{n-1}, \psi_n, \psi_{n+1}$, with solution $\tilde{\psi}_n^+$, $\tilde{\psi}_n^-$. Without YSR impurities, the vortex core LDOS obtained is always axially symmetric, as found previously. There are slight, axially symmetric electron-hole variations at the Fermi wavelength $\lambda_F$ scale, which are smeared out due to dephasing except in the quantum limit. The vortex core LDOS with YSR impurities, obtained from $\tilde{\psi}_n^+$, $\tilde{\psi}_n^-$, is, however, axially asymmetric. The asymmetry is due to the spatial shift in the perturbed CdGM states $\tilde{\psi}_n^+$, $\tilde{\psi}_n^-$ and is induced by the mixing between adjacent CdGM levels ($n+1$ and $n-1$). This asymmetry is roughly given by $\frac{\delta G(\mathbf{r},V)}{G_0} \propto |\tilde{\psi}_n^+|^2 - |\tilde{\psi}_n^-|^2 \propto \pm J^2 e^{-4r_p/\xi_V} \cos(\theta - \theta_p)$, where $\theta$ is the polar angle with respect to the vortex center, $r_p$ and $\theta_p$ provide the length and the angle of the line joining the vortex center and the impurity position and the $\pm$ sign depends on whether the effective mass is negative or positive, i.e., whether the bands have a hole or an electron like character. The magnitude of the perturbation decays exponentially with the distance from the impurity to the vortex center.

As we show in Fig. 4, the observed LDOS asymmetry can be qualitatively reproduced using our theory (Fig. 4c, f). For that purpose we introduce an impurity distribution corresponding to the one in the experiments and add the contribution of each impurity to the asymmetry. Furthermore, we use an isotropic gap for 2H-NbSe$_{1.8}$S$_{0.2}$ and a sixfold anisotropic gap for 2H-NbSe$_2$. Detailed parameters of the calculation are provided in the Supplementary Note 1. Here, we highlight that the exchange coupling $J$ is negative, corresponding to the antiferromagnetic

exchange found in Fig. 3a, b and that we can use the same value for all the impurities. In practice, due to the already mentioned distance dependence, the asymmetry $\frac{\delta G(\mathbf{r},V)}{G_0}$ is however dominated by the few impurities, which are closest to the vortex core.

Let us note that vortices in 2H-NbSe$_2$, with their characteristic strong six-fold star shape, present a rather involved shape of the asymmetry (Fig. 4f). This suggests that the spatial extension of CdGM states determines the overall shape of the asymmetry.

In all, we conclude from our combined theoretical and experimental work that YSR states produce electron-hole asymmetric vortex cores. The YSR states allow visualizing the discrete nature of CdGM levels and their electron-hole asymmetry is translated to large scales. Our theory also suggests that a superconductor with predominantly electron band character should lead to an opposite shift in the LDOS. For example, vortices have been observed in $\beta - \mathrm{Bi_2Pd}$[31–33], which has predominantly electron character[34,35]. YSR states in $\beta - \mathrm{Bi_2Pd}$ have been observed[32] but their influence on CdGM states has not yet been addressed.

## Discussion
Bound states in vortex cores have been considered in the past mostly to address the influence of pair potential disturbances on vortex pinning[36]. Recent calculations find vortex core states induced by magnetic or nonmagnetic impurities, which result in small modifications of the superconducting gap parameter[37]. However, in this work we find that vortex positions and gap parameter do not exhibit visible changes at energies away from the gap edge in presence of magnetic impurities.

Unconventional d-wave, p-wave, f-wave or s ± super-conductors often show pair breaking at atomic impurities and vortices at the same time[38–41]. Our results suggest that vortex bound states might be strongly influenced by pair breaking atomic size impurities. One can envisage experiments using atomic manipulation or deposition to place impurities at certain positions. The position of vortices is easily modified by changing the magnetic field. Groups of geometrically arranged YSR impurities, as those made in refs. [7–9] should lead to significant spatial distortions of the LDOS of vortex states and help identifying new or unconventional bound states.

## Methods

To produce the YSR states in 2H-NbSe$_2$ and 2H-NbSe$_{1.8}$S$_{0.2}$ we introduce Fe impurities during sample growth (about 150 ppm), as identified after the experiment using inductively coupled plasma atomic analysis. In this diluted regime Fe impurities produce practically no changes in the residual resistivity or $T_c$ both for 2H-NbSe$_{1.8}$S$_{0.2}$ or 2H-NbSe$_2$. The amount of Fe impurities is sufficiently small as to leave the superconducting gap and vortex structure unaffected, but large enough to be easily detected in the area occupied by a single vortex. We use a scanning tunneling microscope (STM) to measure the LDOS as a function of the position at 0.8 K. Samples are cleaved at or below liquid He, to allow for a clean atomically flat surface and the tip is prepared in-situ[42].

## Data availability

The data that support the findings of this study are available from the corresponding author upon reasonable request.

## Code availability

The code that supports the findings of this study is available from the corresponding author upon reasonable request.

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

## Acknowledgements

This work was supported by the EU (ERC-StG-679080, ERC-2015-AdG-694097, ERC AdG Mol-2D 788222 and H2020-MSCAIF-2016-751047), by FET-OPEN program (grant 862893 FATMOLS), by EU program Cost (NANOCOHYBRI CA16218 and MOLSPIN CA15128), by Graphene Flagship (grant agreement number 881603, GrapheneCore3 project), by the Spanish State Research Agency (FIS2017-84330-R, RYC2014-15093, MDM-2014-0377, MDM-2015-0538, CEX2018 -000805-M, MAT2017-89993-R co-financed by FEDER and CEX2019-000919-M and PID2020-114071RB-I00/AEI/10.13059/501100011033), by the Deutsche Forschungsgemeinschaft (DFG) under under Germany's Excellence Strategy— Cluster of Excellence Advanced Imaging of Matter (AIM) EXC 2056—390715994, RTG 1995 and GRK 2247, by the Comunidad de Madrid through program NANOMAGCOST-CM (Program No.S2018/NMT-4321) and by the Generalitat Valenciana (Grupos Consolidados IT1249-19, Prometeo Programme,

iDiFEDER/2018/061 and CDEIGENT/2019/022). J.J.B. acknowledges the Marie Curie Fellowship program (H2020-MSCA-IF-2016-751047). A.R. acknowledges support by the MPI-New York City Center for Non-Equilibrium Quantum Phenomena. H.S., E.H., and I.G. acknowledge SEGAINVEX at UAM for design and construction of STM cryogenic equipment. S.P. acknowledges Banco Santander - María de Maeztu program (ref. 102I0112).

## Author contributions

S.P. made the calculations on superconductivity and modeled the system, J.J.B. made the DFT calculations, and V.B. and A.F. carried out the STM experiments. F.M. and M.G.H. made the magnetic characterization. S.M.V. and E.C. made and characterized the samples. J.J.B. interpreted the DFT calculations with the supervision of A.R.. V.B. analyzed and interpreted the STM data with the supervision of E.H. and I.G. The study was devised by S.P., A.L.Y., I.G., and H.S. Manuscript was written by S.P., A.L.Y., and H.S., with contributions from all authors.

## Competing interests

The authors declare no competing interests.
