## [Peer Review File · Nature Communications]

REVIEWER COMMENTS

Reviewer #1 (Remarks to the Author):

The authors investigate the effect of the Yu-Shiba-Rusinov (YSR) bound states of magnetic impurities on the Caroli-de Gennes-Matricone (CdGM) vortex bound states in Fe-doped 2H-NbSe₂ and NbSe_{1.8}SO₂, using scanning tunneling microscope (STM) along with theoretical simulations. The key findings are follows: (1) The coupling between the adjacent CdGM states induced by the YSR states gives rise to the spatial shift of the CdGM states. (2) The directions of the spatial shift in positive and negative energies are opposite to each other, giving rise to the “axial asymmetry” of the difference between the positive and negative energy states. (3) The sign of the observed shifts depends on whether the band character of the superconductor at the Fermi level is electron like or hole like. These findings are novel and provide important insight into the understanding of the relation between the impurities and vortex. And, their experimental and theoretical results support their conclusion. I recommend the publication after revision associated with following question and comments.

(1) According to the authors, the observed spatial shifts of the CdGM states comes from the coupling between the n -th and $n+1$ th quantized CdGM bound states. But both the experimental and theoretical simulations have been conducted at a higher temperature than that of the quantum limit. Since the discretized bound states are completely washed out in their calculation conditions, we can not see how each bound state is affected by the presence of the magnetic impurities. It is better to append the theoretical simulations in the quantum limit. This will make it easy for the readers to understand what happens.

(2) In Fig. 1b and c, the authors indicate spatial shifts in vortex bound states bound states using two dimensional maps of the bound states. However, it is not easy to see the shifts in the two dimensional maps. It is better to append the line profiles of the corresponding LDOS.

(3) Why do the authors choose 0.2 meV in the comparison between the δG images of the experiments and simulations, even though they indicate the energy dependence of the δG maps in Fig. S2d and Fig. S9? If the scenario suggested by the authors is correct, the simulations should reproduce the experimental data at all energies in Fig. S2d and Fig. S9. The authors should provide simulations at all energies.

(4) The key idea to explain the experimental observation is the coupling or mixing between adjacent the CdGM states. However, considering very small energy separation (less than 1 μ eV) between adjacent CdGM states, there should be a few hundred number of CdGM states within their experimental resolution $\sim 300 \mu$ eV at 0.8 K. I cannot understand why the discussion associated with the discrete CdGM states is applicable to explain the experimental data.

Reviewer #2 (Remarks to the Author):

The authors investigated the bound states in superconductors with the combined contributions by the vortex core states, namely the Caroli-de Gennes-Matricone states and that by the magnetic impurities (for s-wave superconductors), namely the Yu-Shiba-Rusinov states. The CdGM states usually exhibit as an electron-hole symmetric feature due to the thermal smearing and/or scattering, while the YSR states are usually asymmetric. They thus studied the vortices in 2H-NbSe₂ and in 2H-NbSe_{1.8}SO₂ with diluted Fe impurities and find the asymmetric spectrum within the vortex core but near the Fe impurity. They have accomplished the study by combining the Hubbard-corrected density functional calculations and experimental observations, and found some kind of consistency. This is systematic work and touches the fundamental issues of these two kinds of bound states in superconductors. While beside seeing this consistency, I am not convinced that this work meets the standard for

publication in Nature Communications due to less-clear originality and novelty, together with the concerns of the measurements of the scanning tunneling microscope on vortices. I list the concerns in the following for consideration.

1. In the STM measurements, generally speaking, it is not easy to observe a perfect symmetric vortex as revealed by mapping the LDOS within the vortex. The symmetric feature can be distorted by many reason, even a defect of slight distortion of the atomic lattice. In the studied materials, the accompanied CDW states also give some influence which can strongly affect the configuration of a vortex. Thus the authors need to prove that these effects (other than the Fe impurities) can be removed and how. In addition, in the calculation, it is also very important how the Fe is put to the atomic lattice, it can locate in the interstitial positions, or substitute to Nb? If possible, one may need to illustrate the precise positions of the Fe sites through a topography measurement, then do more elegant calculations. This is requested because these details do give influence on the mapping of LDOS within the vortex.

2. The authors need to convince that the concerned issue has its strong impact to the community. Reading from the manuscript, I get a feeling that the asymmetric pattern obtained by the difference conductivity with the subtraction of the LDOS at positive and negative energies is attributed to the YSR states of impurity effect within the vortex core, which can be simulated by the already-known DFT calculations. The consistency seems to be there, and the results may show a way for the study of coupled superconducting bound states, but what is the significance? This point is not clear to this reviewer and the authors may elaborate more.

Reviewer #1: *The authors investigate the effect of the Yu-Shiba-Rusinov (YSR) bound states of magnetic impurities on the Calori-de Gennes-Matricone (CdGM) vortex bound states in Fe-doped 2H-NbSe₂ and NbSe_{1.8}SO₂, using scanning tunneling microscope (STM) along with theoretical simulations. The key findings are follows: (1) The coupling between the adjacent CdGM states induced by the YSR states gives rise to the spatial shift of the CdGM states. (2) The directions of the spatial shift in positive and negative energies are opposite to each other, giving rise to the “axial asymmetry” of the difference between the positive and negative energy states. (3) The sign of the observed shifts depends on whether the band character of the superconductor at the Fermi level is electron like or hole like. These findings are novel and provide important insight into the understanding of the relation between the impurities and vortex. And, their experimental and theoretical results support their conclusion.*

I recommend the publication after revision associated with following question and comments.

Answer: We acknowledge the Reviewer for these comments and for supporting the publication of our work. The Reviewer summarizes carefully our main findings, showing that we find a new coupling between bound states in superconductors, which manifests as a shift in the CdGM states leading to electron-hole and axially asymmetric vortex core density of states. This is found in experiment and theory. Remarkably, theory includes very careful calculations taking into account the actual distribution of impurities found in the experiment.

Reviewer #1: *(1) According to the authors, the observed spatial shifts of the CdGM states comes from the coupling between the n -th and $n+1$ th quantized CdGM bound states. But both the experimental and theoretical simulations have been conducted at a higher temperature than that of the quantum limit. Since the discretized bound states are completely washed out in their calculation conditions, we can not see how each bound state is affected by the presence of the magnetic impurities. It is better to append the theoretical simulations in the quantum limit. This will make it easy for the readers to understand what happens.*

Answer: We thank the Reviewer for this suggestion. Following the Reviewer, we provide now the calculations of the tunneling conductance in the quantum limit. We provide the results and describe all details in the supplementary information.

For completeness, we append the result below in a figure and provide here a summarized description and discussion.

To obtain these results, we have considered a single impurity close to the vortex center. We have calculated the difference between positive and negative bias voltages using perturbation theory. In Fig.S1a-c below we show results in the quantum limit. In Fig.S1a we show the derivative of the Fermi function (red) and the CdGM levels (green lines). To reach the quantum limit, we use a low temperature, of 100 mK and a very large Fermi wavelength, of 10 nm. We see that temperature broadening is smaller or of order of level separation. In this situation, we find the vortex core shape shown in Fig.S1b. We see that the vortex core is nearly totally symmetric. In Fig.1c we show the actual density of states as a function of the distance from the vortex center along the line between the impurity and the vortex center (blue) and along the opposite direction, from the vortex center (red). There is a slight asymmetry, which is however very weak in the tunneling conductance (Fig.S1b). However, when we consider a Fermi wavelength of 1 nm (Fig.S1d-f), the temperature induced broadening is considerably larger than the level separation (Fig.S1d). Here we observe a core that is considerably asymmetric, when making the difference between electron and hole states (Fig.S1d). There is a significant asymmetry with respect to the position of the impurity when plotting the density of states along the same lines as above (Fig.S1f). The fast oscillations at the Fermi wavelength are strongly damped and the dominant

effect now is the coupling to the YSR state. Thus, as we mentioned in the manuscript, the coupling is small as compared to the oscillations at the Fermi wavelength. However, it is significant and has a considerable influence on the experiment.

FIG. S1. **Quantum limit vs thermal broadening.** **a** Vortex bound states are shown as green lines (spin up states as solid lines and spin down states as dashed lines). The derivative of the Fermi function is shown in red. We consider a Fermi wavelength $\lambda_F = 10$ nm, a temperature of 100 mK and a vortex size $\xi_V = 30$ nm. The resulting difference between the tunneling density of states at positive and negative bias voltages (at $V = \pm 0.2$ mV) is shown in **b**, with an impurity located at 20 nm from the vortex center (black dot). The color scale is shown by the bar at the right. We observe the oscillatory electron-hole asymmetry that is characteristic of vortex bound states. In **c** we trace in blue the difference between electron and hole states along a line from the vortex center towards the impurity (blue dashed line in **b**) and in red along the opposite direction (red dashed line in **b**). In **d-e** we show the same results, with the same parameters, except that we take $\lambda_F = 1$ nm.

Reviewer #1: (2) In Fig. 1b and c, the authors indicate spatial shifts in vortex bound states using two dimensional maps of the bound states. However, it is not easy to see the shifts in the two dimensional maps. It is better to append the line profiles of the corresponding LDOS.

Answer: We thank the Reviewer very much for this suggestion. The new figure one includes the line profiles (new Figure 1d). We copy it here:

FIG. 1. **Interaction between CdGM and YSR states.** **a** The YSR state is schematically represented by the red and blue dots with arrows representing the spin. The value of the superconducting gap as a function of the radius is shown in green for energies above and below the Fermi level. The CdGM states are schematically shown by colored circles. The difference in energy between levels, Δ^2/E_F , with Δ the superconducting gap and E_F the Fermi energy, is strongly exaggerated. Green arrow on top schematically represents the vortex core size ξ_v . The interaction between CdGM and YSR bound states leads to a spatial shift in the CdGM states, which is different for electron and hole components of the LDOS inside the superconducting gap, $N(E > 0)$ (**b**) and $N(E < 0)$ (**c**). In **d** we show line profiles of the images in **b** (blue-green) and of **c** (magenta). We also show a profile of their difference in orange. In **e** we show the difference image between **b** and **c**. The LDOS is larger along the direction of the YSR impurity. This occurs for a hole band and a single impurity. For an electron band, we expect the opposite result. The position of the YSR impurity is represented by a black dot and the vortex core center by the crossing point between black lines in **b**, **c**, **d** and **e**.

Reviewer #1: (3) Why do the authors choose 0.2 meV in the comparison between the δG images of the experiments and simulations, even though they indicate the energy dependence of the δG maps in Fig. S2d and Fig. S9? If the scenario suggested by the authors is correct, the simulations should reproduce the experimental data at all energies in Fig. S2d and Fig. S9. The authors should provide simulations at all energies.

Answer: We thank the Reviewer for making this point. The simulations reproduce the whole bias dependence, at least qualitatively. We choose to make a more quantitative comparison at an energy range where the asymmetry is large. Going down to zero bias makes no sense, as any electron-hole anisotropy vanishes at the Fermi energy. Approaching the gap leads us away from the approximation made in the calculations ($E \ll \Delta$, with Δ the superconducting gap). We have made a more detailed comparison and added it to the supplement. We include this comparison here:

FIG. S12. **Bias voltage dependence of $\frac{\delta G(r, V)}{G_0}$ in 2H-NbSe_{1.8}S_{0.2}, compared to our calculations.** We show $\frac{\delta G(r, V)}{G_0} = \frac{G(r, V) - G(r, -V)}{G_0}$ for the bias voltages marked in each panel, obtained from calculations (upper panels) and experiment (lower panels). Black dots provide the position of magnetic impurities. Color scale is given by the bars on the right.

Reviewer #1: (4) *The key idea to explain the experimental observation is the coupling or mixing between adjacent the CdGM states. However, considering very small energy separation (less than 1 μeV) between adjacent CdGM states, there should be a few hundred number of CdGM states within their experimental resolution $\sim 300 \mu\text{eV}$ at 0.8 K. I cannot understand why the discussion associated with the discrete CdGM states is applicable to explain the experimental data.*

Answer: The Reviewer is right that this issue needs to be better clarified. With the new figures and the text added to the manuscript and the supplement, we believe that this is now clearer. As mentioned above, the component varying over the vortex core size $\xi_v=30\text{nm}$, which highlights the coupling between YSR and CdGM states, remains visible at high temperatures because it accumulates over CdGM states without being destroyed by temperature smearing. On the contrary, the oscillating component at the Fermi wavelength $\lambda_F=1 \text{ nm}$ is smeared by the temperature induced overlap between states.

Reviewer #2: *The authors investigated the bound states in superconductors with the combined contributions by the vortex core states, namely the Caroli-de Gennes-Matricon states and that by the magnetic impurities (for s-wave superconductors) , namely the Yu-Shiba-Rusinov states. The CdGM states usually exhibit as an electron-hole symmetric feature due to the thermal smearing and/or scattering, while the YSR states are usually asymmetric. They thus studied the vortices in 2H-NbSe2 and in 2H-NbSe1.8S0.2 with diluted Fe impurities and find the asymmetric spectrum within the vortex core but near the Fe impurity. They have accomplished the study by combining the Hubbard-corrected density functional calculations and experimental observations, and found some kind of consistency. This is systematic work and touches the fundamental issues of these two kinds of bound states in superconductors.*

Answer: We thank the Reviewer very much for these comments, particularly for emphasizing the fundamental aspect of our work and how it addresses systematically an interesting problem. The coupling between bound states in superconductors, while retaining quantum coherence, is important for quantum manipulation of bound states. We present here a novel transfer mechanism of the quantum property of the electron-hole asymmetry from a few-nm sized YSR states into the CdGM states extended several tens of nm far from the quantum limit. As the Reviewer already acknowledged, this "touches the fundamental issues" and, to our best knowledge, this is the first report of the asymmetry of a vortex at high temperatures.

We would like to stress that the asymmetric pattern we observed is not a simple extension of the asymmetry of the YSR states projected into the CdGM states. In addition to the thermal smearing, a huge number of CdGM states participate in our tunneling conductance measurements. Each state spatially oscillates with the length scale of order of 1 nm. These oscillations are washed out by temperature in absence of coupling to magnetic impurities. But the coupling to YSR remains, suggesting a counterintuitive revival of the quantum behavior CdGM states.

Reviewer #2: *While beside seeing this consistency, I am not convinced that this work meets the standard for publication in Nature Communications due to less-clear originality and novelty, together with the concerns of the measurements of the scanning tunneling microscope on vortices. I list the concerns in the following for consideration.*

1. In the STM measurements, generally speaking, it is not easy to observe a perfect symmetric vortex as revealed by mapping the LDOS within the vortex. The symmetric feature can be distorted by many reason, even a defect of slight distortion of the atomic lattice. In the studied materials, the accompanied CDW states also give some influence which can strongly affect the configuration of a vortex. Thus the authors need to prove that these effects (other than the Fe impurities) can be removed and how.

Answer: We thank the Reviewer for placing this concern. As we can see in the figure S3a, also reproduced below, vortices in NbSe2 are symmetric. The influence of an atomic size defect hardly affects vortices in these materials, because the vortex is much larger than any interatomic distances. Of course, materials with a smaller coherence length (such as the Fe pnictides or the cuprates) might be much more sensitive to atomic size features. But this is not the case in NbSe2. Furthermore, we would like to draw the attention to the Reviewer that the calculations have been made by placing all impurities at fixed positions and the shape found in the calculations coincides precisely with what we find in the experiment. For example, the figures 4b and 4e of the manuscript present vortices that have asymmetric shapes on opposite directions. This is not a coincidence and cannot be caused by slight differences in the local electronic properties of the samples. We have included a new figure in the supplement, where we show three different vortices in 2H-NbSe1.8S0.2 showing asymmetries along three different directions. The charge

density wave cannot cause such a behavior. Finally, the electron-hole asymmetry of the conductance does not depend on the potential K , but it depends solely on J .

Figures S3 a and c: We show the zero bias conductance at a vortex in 2H-NbSe₂ in absence of magnetic impurities in **a**. The difference between positive and negative bias is shown in **c**. Notice that there is no asymmetry. As shown in the Figure 4 **d** of the main text, a similar vortex in a sample with Fe impurities, shows a strong asymmetry.

Reviewer #2: *In addition, in the calculation, it is also very important how the Fe is put to the atomic lattice, it can locate in the interstitial positions, or substitute to Nb? If possible, one may need to illustrate the precise positions of the Fe sites through a topography measurement, then do more elegant calculations. This is requested because these details do give influence on the mapping of LDOS within the vortex.*

Answer: We thank the Reviewer for making this important comment. The topography provides the Se lattice. We do not find clear protrusions or indications that there are Fe atoms on top of the Se lattice. Furthermore, we have made calculations of a relaxed Fe atom on top of the Se surface, finding a strongly decreased exchange interaction. We have added this figure into the Supplement and provide it also below.

FIG. S8. Fe located on top of the Se surface. Lateral (**a**) view and view from the top (**b**) of the spin density of a Fe atom located on top of the Se surface in 2H-NbSe₂ and in 2H-NbSe_{1.8}S_{0.2} (**c**, **d**). Se atoms are shown in orange, Nb atoms in ochre and S atoms in yellow. We plot the spin isosurface corresponding to a spin imbalance by 0.002 in red (spin up) and blue (spin down).

Even if the surface layer is Se, we can estimate the position of the Fe atom causing the YSR state. Thus, we have also added a figure where we identify, atom by atom the position of the Fe impurity, as requested by the Reviewer.

FIG. S9. **Atomic size map around a Fe impurity in $2\text{H-NbSe}_{1.8}\text{S}_{0.2}$.** In **a** we show a topographic image taken in $2\text{H-NbSe}_{1.8}\text{S}_{0.2}$ with a bias voltage of 5 mV and a tunneling current of 0.1 nA. In **b** we show a close up view of the area marked by a white rectangle in **a**. The Se atomic lattice is marked by green dots. Atomic directions are marked by white arrows. We mark the position of the Fe impurity by a red dot. **c** Tunneling conductance map at a bias voltage of 0.2 mV. Color scale given by the right bar (we plot the tunneling conductance normalized at bias voltages above the superconducting gap). The position of the Fe impurity is marked by a red dot.

Reviewer #2: 2. *The authors need to convince that the concerned issue has its strong impact to the community. Reading from the manuscript, I get a feeling that the asymmetric pattern obtained by the difference conductivity with the subtraction of the LDOS at positive and negative energies is attributed to the YSR states of impurity effect within the vortex core, which can be simulated by the already-known DFT calculations. The consistency seems to be there, and the results may show a way for the study of coupled superconducting bound states, but what is the significance? This point is not clear to this reviewer and the authors may elaborate more.*

Answer: We thank the Reviewer for helping us to improve the impact of our paper. The calculations leading to the electron-hole asymmetry are completely new and the results were not predictable without doing such calculations. It was a great surprise to us that we could establish a connection between the two kinds of bound states that can be found in superconductors. This discovery is far reaching. It significantly contributes to our understanding of tunneling in superconductors and allows establishing a line of work that can be very helpful to identify exotic forms of superconductivity. But, for us it has been a great success to find a new effect (asymmetric vortex cores) and achieve a full explanation of this effect, with new and far reaching insight. We have added a sentence in the abstract and in the conclusions to indicate the use of the effect we have discovered in the identification of exotic superconducting phases.

REVIEWERS' COMMENTS

Reviewer #1 (Remarks to the Author):

First of all, I would like to express my deep appreciation for the courteous author's response. All of the author's answers to my questions and comments are satisfactory and meet my requirements. Therefore, I believe that this paper is ready for publication in Nature Communications.

Reviewer #2 (Remarks to the Author):

I find that most of my concerns/suggestions have been well responded, and they provide more data and explanations which allow me to make positive recommendation. This paper covers a combined investigation of experiments and theoretical efforts and tackles fundamental issue of the joint contribution by impurity bound states and the vortex bound states. I think it is acceptable for publication in Nature Communications.

REVIEWERS' COMMENTS

Reviewer #1 (Remarks to the Author):

First of all, I would like to express my deep appreciation for the courteous author's response. All of the author's answers to my questions and comments are satisfactory and meet my requirements. Therefore, I believe that this paper is ready for publication in Nautre Communications.

Answer: We thank the Referee for the positive comments on our paper and our answers.

Reviewer #2 (Remarks to the Author):

I find that most of my concerns/suggestions have been well responded, and they provide more data and explanations which allow me to make positive recommendation. This paper covers a combined investigation of experiments and theoretical efforts and tackles fundamental issue of the joint contribution by impurity bound states and the vortex bound states. I think it is acceptable for publication in Nature Communications.

Answer: We thank the Referee for the positive recommendation and for summarizing our work.